# Comparative Immunoreactivity Analyses of Hantaan Virus Glycoprotein-Derived MHC-I Epitopes in Vaccination

**DOI:** 10.3390/vaccines10040564

**Published:** 2022-04-06

**Authors:** Baozeng Sun, Junqi Zhang, Jiawei Wang, Yang Liu, Hao Sun, Zhenhua Lu, Longyu Chen, Xushen Ding, Jingyu Pan, Chenchen Hu, Shuya Yang, Dongbo Jiang, Kun Yang

**Affiliations:** 1Department of Immunology, Basic Medicine School, Air-Force Medical University (the Fourth Military Medical University), Xi’an 710032, China; sbz010115@163.com (B.S.); zjq000211@163.com (J.Z.); vv_jevin@163.com (J.W.); liuyangyang9610@163.com (Y.L.); sunhao.98721.com@163.com (H.S.); ruzzi618@163.com (Z.L.); chenlongyu410@163.com (L.C.); dxs001630@163.com (X.D.); jy648700471@163.com (J.P.); 18579121005@163.com (C.H.); yangshuxiaoya@163.com (S.Y.); 2Shaanxi Provincial Center for Disease Control and Prevention, Xi’an 710054, China; 3Tangshan Sannvhe Airport, Tangshan 063000, China; 4Department of Epidemiology, Public Health School, Air-Force Medical University (the Fourth Military Medical University), Xi’an 710032, China

**Keywords:** pan-MHC-I, Hantaan virus (HTNV), glycoprotein (GP), immunoreactivity, comparative immunology

## Abstract

MHC-I antigen processes and presentation trigger host-specific anti-viral cellular responses during infection, in which epitope-recognizing cytotoxic T lymphocytes eliminate infected cells and contribute to viral clearance through a cytolytic killing effect. In this study, Hantaan virus (HTNV) GP-derived 9-mer dominant epitopes were obtained with high affinity to major HLA-I and H-2 superfamilies. Further immunogenicity and conservation analyses selected 11 promising candidates, and molecule docking (MD) was then simulated with the corresponding MHC-I alleles. Two-way hierarchical clustering revealed the interactions between GP peptides and MHC-I haplotypes. Briefly, epitope hotspots sharing good affinity to a wide spectrum of MHC-I molecules highlighted the biomedical practice for vaccination, and haplotype clusters represented the similarities among individuals during T-cell response establishment. Cross-validation proved the patterns observed through both MD simulation and public data integration. Lastly, 148 HTNV variants yielded six types of major amino acid residue replacements involving four in nine hotspots, which minimally influenced the general potential of MHC-I superfamily presentation. Altogether, our work comprehensively evaluates the pan-MHC-I immunoreactivity of HTNV GP through a state-of-the-art workflow in light of comparative immunology, acknowledges present discoveries, and offers guidance for ongoing HTNV vaccine pursuit.

## 1. Introduction

Hemorrhagic fever with renal syndrome (HFRS) is a viral zoonosis. It is caused by Old World hantaviruses, including Hantaan virus (HTNV), Dobra–Belgrade virus (DOBV), Puumala virus (PUUV), and Seoul virus (SEOV). In China, HTNV is a common pathogenic species causing severe HFRS disease. Currently, this disease occurs globally in more than 70 countries, and an approximate 70~90% notification was reported in China, where HFRS is still considered a serious public health problem because it is highly endemic, with about 20,000–50,000 incident cases per year, leading to a fatality rate of around 3~10% [1]. The goal is to eliminate the HFRS epidemic in the next 10 years, considering the forecasts of around 90,120 cases [2].

The HTNV genome consists of S, M, and L segments, which encode nucleocapsid protein (NP), Gn and Gc glycoprotein (GP), and RNA-dependent RNA polymerase (RdRp) [3], respectively. The structural proteins NP and GP play important roles in evoking the humoral and cellular immune responses against HTNV [4] and are responsible for strong immune responses. Hence, gene therapy against the two structural proteins is considered to be an effective treatment for HFRS caused by hantaviral infection. Meanwhile, protective immunity elicited through infection with recombinant HTNV glycoprotein in mouse models has also been demonstrated [5]. The HTNV glycoprotein could also be a potent immunogen to induce T-cell responses [6].

In recent years, HTNV GP-derived MHC class I (MHC-I)-restricted epitopes have been recorded. Ma et al. identified the restricted CTL epitope GP6 aa456–aa463 (ITSLFSLL) in C57BL/6 mice with a mouse MHC-I genotype H2-Kb that could be used in the design of a vaccine against HTNV infection [7]. Tang et al., on the other hand, demonstrated that seven HLA-A0201-restricted HTNV GP-specific epitopes induced protective CTL responses after HTNV infection in patients with milder HFRS disease. Meanwhile, transgenic mice pre-inoculated with three of these epitopes (VV9 (aa8–aa16, VMASLVWPV), SL9 (aa 996–aa 1004, SLTECPTFL), and LL9 (aa 358–aa 366, LIWTGMIDL)) revealed that LL9 functioned as an immunodominant protective epitope [8]. Nevertheless, the MHC-I-restricted HTNV GP epitopes were proven to be immunogenic and protective in a limited immunogenetic background, but there remains a lack of a comprehensive understanding.

In this study, the HTNV GP-derived MHC-I dominant epitopes of mouse H2 and HLA-I superfamilies covering 99.3% of the population [9] were predicted by combining five authentic methods. Highly conserved, well-immunogenic selective candidates were obtained and cross-validated by molecular docking simulations. Hierarchical clustering illustrated pan-MHC immunoreactive hotspots on HTNV GP and epitope-recognizing MHC-I similarity across allogeneic and species. We investigated HTNV CTL epitopes for immune affinity, immunogenicity, conservation, and molecular docking. The cross-reactivity of HTNV CTL epitopes was investigated by hierarchical clustering. The multidimensional exploration laid a theoretical and technical foundation for the development of protective CTL epitope vaccines that can activate HTNV-specific population immunity in the context of pan-MHC immunology.

## 2. Materials and Methods

### 2.1. HTNV GP Protein Sequences Retrieval

The glycoprotein (GP, accession no: KT885048.1) of HTNV 76-118 was obtained from NCBI GenBank as the input for sequential in silico analyses. To analyze the variant amino acid sites and their influences on the affinity differences of GP-dominant epitopes among HTNV strains, the protein sequences of the reported isolates (148 envelope glycoproteins shown in Appendix A) were obtained from NCBI GenBank.

### 2.2. HTNV GP Pan-MHC-I Epitopes Prediction and Screening

To obtain high-affinity epitope candidate peptides as comprehensively and unbiasedly as possible, we utilized a variety of prediction algorithms to perform sequential oligopeptide segmentation of the target GP sequence and affinity calculation among MHC-I molecules. The MHC-I molecules included 6 mouse H2 genotypes of H2-Db, H2-Dd, H2-Kb, H2-Kd, H2-Kk, and H2-Ld, and 9 human HLA-I superfamilies, including HLA-A1 (A0101, A2601, A3001, and A3002), HLA-A2 (A0201, A0203, A0206, and A6802), HLA-A3 (A0301, A1101, A3001, A3101, A3301, and A6801), HLA-A24 (A2301, A2402), HLA-A3201, HLA-B7 (B0702, B3501, B5101, and B5301), HLA-B8 (B0801), HLA-B15 (B1501), HLA-B44 (B4001, B4402, and B4403), and HLA-B58 (B5701 and B5801). For each MHC-I molecule, the affinities of 9-mer GP peptides were predicted using algorithms, such as IEDB-recommended (http://tools.iedb.org/mhcii/, accessed on 6 August 2021) [10], SMMPMBEC (http://tools.immuneepitope.org/mhci/, accessed on 6 August 2021) [11], NetMHCpan4.1 (http://www.cbs.dtu.dk/services/NetMHCpan/, accessed on 6 August 2021) [12], SYFPEITHI (http://www.syfpeithi.de/bin/MHCServer.dll/EpitopePrediction.htm, accessed on 6 August 2021) [13], and Rankpep (http://imed.med.ucm.es/Tools/rankpep.html, accessed on 6 August 2021) [14,15], and the predicted epitopes with accounting scores in the top 2% of each algorithm’s result were the candidate dominant epitopes. We then chose the epitopes that appeared in at least three prediction algorithm results and regarded them as the dominant epitopes.

### 2.3. Conservation Analysis

To determine the degree of evolutionary conservation of the dominant antigenic epitope among the viral species sequences, we used the BLASTP (https://blast.ncbi.nlm.nih.gov/Blast.cgi, accessed on 21 November 2021) tool for conservation analysis of the predicted high-affinity 9-mer peptides. Among them, the evaluation criterion for interspecies conservation was Orthohantavirus (taxid: 1980442), excepting Hantaan hantavirus (taxid: 1980471). The intraspecific conservative evaluation criterion was Hantaan hantavirus (taxid: 1980471), other than Hantaan virus (strain 76-118) (taxid: 11602). In the analysis results, the conservative E-value was <10^−5^ and the conservative peptide sequences between HTNV and human (taxid: 9606) or mouse (taxid: 10088) were excluded simultaneously. The dominant epitopes could therefore be classified into four categories based on conservation status: interspecies- intraspecific- interspecies- intraspecific+, interspecies+ intraspecific-, and interspecies+ intraspecific+. “+” means that the epitopes were conservative and “-” means that the epitopes were not conservative.

### 2.4. Immunogenicity Analysis

Peptides with high affinity may not sufficiently induce immune responses [16]. In addition to immunoreactivity, the antigen should also be immunogenic. The immunogenicity is determined by the amino acid sequence [17]. VaxiJen 2.0 (http://www.ddg-pharmfac.net/vaxijen/VaxiJen/VaxiJen.html, accessed on 12 August 2021) [18] was used to calculate the immunogenicity of the 9-peptide epitope. Peptides were considered immunogenic by a probability score of >0.5 as the positive criterion; otherwise, they were not considered immunogenic.

### 2.5. Docking of pMHC-I Molecules

After the above serial analysis, we obtained high-affinity, evolutionarily conserved, and immunogenic dominant epitopes called selective epitopes. Next, we used the peptide sequences to perform docking simulations with the molecular conformations of each MHC-I isoform to obtain a series of molecular docking thresholds for the pan-MHC-I dominant epitopes. The structural data of each MHC-I allele (HLA-A1 (HLA-A0206 (3OXR), HLA-A0101 (4NQV)), HLA-B7 (HLA-B0702 (5EO1), HLA-B3501 (1A9E), HLA-B5101 (1E28), HLA-B5301 (1A1N)), HLA-B8 (HLA-B0801 (4QRP)), HLA-B15 (HLA-B1501 (1XR9)), HLA-B44 (HLA-B4402 (3KPL)), H2-Ld (6L9M), H2-Kb (6JQ3), and H2-Db (1JUF)) were deposited in the RCSB PDB database (https://www.rcsb.org/, accessed on 18 August 2021).

Docking of pMHC-I was performed using HPEPDOCK (http://huanglab.phys.hust.edu.cn/hpepdock/, accessed on 12 December 2021) [19]. The docking model was obtained by inputting the 9-mer dominant epitope sequence with the MHC-I molecule PDB format file. Each docking assay yielded 100 simulated structures, and the top 10 were selected as high-confidence docking results.

### 2.6. HTNV GP Peptides and Pan-MHC-I Clustering

Polymorphisms in the MHC-I molecules and the diversity of the amino acid sequences of the epitope peptides formed the interaction between their two groups. In order to visualize the relationship between them, the affinity index of the MHC-I superfamily and the HTNV GP-related peptide was subjected to two-way hierarchical clustering using TBtools [20]. After the affinity ranking data were processed by base 2 logarithm and Z-Score minus, we used the Complete Method to perform the two-way hierarchical clustering by Euclidean distance. The analysis showed that the higher the score, the stronger the affinity of the peptide to MHC-I molecules. The analysis contained 33 pan-MHC-I molecules interacting with 1126 HTNV GP epitopes and the heatmap was plotted for representation.

### 2.7. Sequence Alignment of HTNV Variants

On the basis of HTNV strain 76-118, we performed multiple sequence alignments of the hotspots with those of 148 variants with ClustalX2.1 (Conway Institute UCD Dublin, Ireland). The alignment results were also analyzed on WebLogo (http://weblogo.berkeley.edu/logo.cgi, accessed on 3 December 2021) [21]. The height of the trait represented the occurrence frequency of the amino acid mutants among the different variants. Based on the alignment results, all 9 relevant peptides (referred to as mutation residues) were further analyzed with HLA-I molecules for affinity changes. TBtools was used to draw the heatmap of the binding affinity delta value between strain 76-118 and the variants of the corresponding HLA-I and 9-mer epitopes, where the binding affinity took the logarithm of base 2. The delta value of %Rank as minus represented the epitope of strain 76-118 with better affinity. On the other hand, plus values indicated those of variants acquiring better binding performance with the corresponding MHC-I molecule.

Afterward, a scatter plot by Origin 2021 (OriginLab, Northampton, MA, USA) visualized the affinity of the strain 76-118 peptide as the abscissa and the variant as the ordinate. Finally, the dominant epitopes of all mutated 9-mer peptides were predicted and analyzed to determine whether amino acid mutations altered the epitope dominance.

### 2.8. Pan-MHC-I-Restricted HTNV GP Epitopes Application by Literature Review

Through previously reported data, applications of HTNV GP-restricted epitopes were summarized. Epitopes that elicited immune responses and protection against HTNV in local patients and animal models were enrolled and then compared with the dominant ones from our study in order to verify the subjectivity of all of the results.

### 2.9. Vaccine, Animals, and Immunization

The bivalent HFRS inactivated vaccine (HANPUWEI^®^), which was produced by the Changchun Institute of Biological Products Co., Ltd., Changchun, China, was used in this study. The vaccine was composed of inactivated and purified HTNV and SEOV in hamster kidney cells (there is no monovalent HFRS vaccine available in China).

Eight-week-old specific-pathogen-free female mice of two kinds were purchased from the Laboratory Animal Centre of the Fourth Military Medical University. The two types of inbred mice included c57 and BALB/c within the MHC-I haplotypes of H2b and H2d, respectively. Six mice of each kind were randomly divided into two groups. The experimental groups were inoculated with a single 50 μL dose of bivalent inactivated vaccine. The mice in the control groups were injected with a single 50 μL dose of sterile PBS (phosphate-buffered solution). The immunized mice were sacrificed 6 days later. An ELISpot assay was used for cellular evaluation. Animals were taken good care of, and all of the experiments were carried out according to the animal experiment guidance.

### 2.10. Peptides and ELISpot Assay

Three of the validated HTNV GP-derived HLA-A2 dominant 9-mer epitopes were acquired from a neighboring research crew [7,22], Single peptides were diluted with 10 μg/mL PBS for the IFN-γ ELISpot assay. Briefly, the IFN-γ-specific capture antibody was diluted with 5 μg/mL (1:250) sterile PBS and placed on coated ELISpot plates overnight at 4 °C. The mice were sacrificed and their spleens were dissected, after which the monocyte suspension was ground. After erythrocyte lysis, the splenocytes were rinsed and re-suspended. Two hours after the ELISpot plates were blocked with RPMI-1640 containing 10% fetal bovine serum at room temperature, 1 × 10^6^ splenocytes were added to each pore and stimulated with a final concentration of 5 μg/mL synthetic GP peptides. The plates were cultured in a 5% CO_2_ incubator at 37 °C for 24 h. Completed medium was used as the negative control. Con A (10 μg/mL) was used as the positive control. After incubation, the culture plates were washed with H_2_O and PBST, and then incubated with 2 mg/mL of relevant biotinylated rat anti-mouse IFN-γ antibody at room temperature for 2 h. After washing with PBST (PBS with 0.05% Tween-20), the plates were incubated with streptavidin-HRP 1:100 for 1 h, and then 3-amino-9-ethylcarbazole (AEC; DAKEWEI, Shenzheng, China) was added to the HRP substrate, and the reaction was stopped by washing with water. After air-drying, we used the CTL ELISpot Reader (CTL, Kennesaw (Atlanta), GA, USA) to count IFN spots generated by the AEC substrate. Each experiment was performed in triplicate, and all of the results are shown as the average value of spot-forming cells (SFCs) per 10^6^ splenocytes.

## 3. Results

### 3.1. HTNV GP Epitopes for Mouse MHC-I and Major HLA-I Supertypes

We performed bioinformatics analysis using the multiple computational tools that were mentioned in the Methods. Table 1 lists the numbers of predicted dominant epitopes in HLA-I, and Table 2 lists the numbers of mouse MHC-I that were generated by each of the prediction tools. We obtained 229 epitopes in HLA-I and 83 epitopes in H2 (specific epitopes are shown in Appendix A). The most comprehensive coverage for HLA-I alleles was NetMHCpan-4.1, and the H2 subtype was best covered by IEDB and NetMHCpan-4.1. Altogether, NetMHCpan-4.1 had the most complete subtype among the MHC-I prediction tools. According to the results of the HLA-I alleles, the number of HLA-A3-restricted dominant epitopes was the highest (57 peptides of full-length GP, Table 1), and the number of H2-Db was the highest in the H2 subtype (18 peptides of full-length GP, Table 2).

Within the ranks of the affinity between HTNV 9-mer peptides and different MHC-I molecules, a heatmap was drawn to show the regional affinity (as shown in Appendix A). The GP of HTNV 76-118-derived 1127 single-residue-advancing 9-mer peptides at the distribution of 1135 amino acids, and the data were taken from NetMHCPan-4.1. The heatmap columns are labeled with MHC-I subtypes and the rows are labeled with epitopes. The graph shows a single gradient. The smaller the %Rank, the darker the red. Generally, the binding strength of epitopes was regionally distributed, ranging from No. 134–162, No. 184–205, No. 212–214, No. 438–456, No. 495–499, No. 791–793, No. 922–926, No. 995–996, and No. 1057–1104 with good affinity. However, No. 759–780, No. 800–815, No. 953–982, No. 988–1010, and No. 1040–1047 across different subtypes showed poor binding ability. The dominant epitope region of HLA-A was the most concentrated, and that of H2 had better coverage than that of the HLA-B subtypes. The overall coverage of good affinity for HTNV GP-derived 9-mer peptides to different MHC-I subtypes was, therefore, HLA-A2 > HLA-A1 > HLA-A3 > HLA-A24 > H2 > HLA-B7 > HLA-B58 > HLA-B15 and HLA-B44. Most of the pan-MHC-I dominant epitopes fell in the nine hotspots of high affinity and underwent the following explorations.

### 3.2. Conservation Status of HTNV GP 9-Mer Dominant Epitopes

To determine the degree of evolutionary conservation of the dominant antigenic epitope among viral species sequences, we used the criteria described in the Methods to perform conservation analysis of the predicted screened high-affinity 9-mer peptides using the BLASTP tool. The statistical results of the conservation analyses for all dominant epitopes are listed in Table 3. The various MHC-I epitopes were classified into four classes (Interspecies- Intraspecies-, Interspecies- Intraspecies+, Interspecies+ Intraspecies-, and Interspecies+ Intraspecies+). The number of conserved HLA-I-restricted dominant epitopes was higher than that of the H2-restricted dominant epitopes. This was the result of HLA-I summarizing the superfamilies, while the identification of H2-restricted epitopes required the approval of three out of five algorithms for six subtypes. At the same time, it was evident that the dominant epitope showed strong intraspecific conservation, but weak interspecific conservation.

### 3.3. Immunogenicity of HTNV GP 9-Mer Peptides

Peptides that adequately induce immune responses require not only high affinity but also immunogenicity. Therefore, we performed immunogenicity analysis of all the 9-mer GP epitopes. The results revealed that 551 of the 1126 HTNV GP 9-mer peptides were immunogenic. Specifically, the dominant epitopes of 124 of the 229 HLA-I and 39 of the 83 H2 subtypes were regarded as immunogenic peptides (Appendix A).

### 3.4. Hierarchical Clustering Showed Interaction between pan-MHC-I Molecules and HTNV 9-Mer Peptides

Through the above analysis, the dominant epitopes with high affinity, evolutionary conservation, and immunogenicity were named “selective” epitopes. However, these promising targets could not reflect the entire picture of HTNV GP peptides being processed by pan-MHC-I supertypes. At the same time, it was previously determined that the MHC-II haplotypes clustered by HTNV GP immunoreactivity represent the similarity between individuals even across species [23]. In order to investigate whether the same phenomenon would occur in the pan-MHC-I-restricted HTNV GP presentation, we subsequently performed hierarchical cluster analysis of 1127 HTNV 9-mer peptides (Figure 1). Thirty-three MHC-I subtypes were assigned to three clusters, including HLA-I-exclusive and two cross-reactive clusters (HLA major and H2 major). In the HLA-I-exclusive cluster, the HLA-A3001 scores were similar to those of HLA-A3 (-A0301, -A1101, -A3101, -A3301, and -A6801), more so than those of other HLA-A1 superfamilies, suggesting an HLA-A3-like characteristic in HTNV GP processing [24]. HLA-A2 (-A0201, -A0203, -A0206, and -A6802) scored similarly. In the HLA-I-exclusive clusters, HLA-B7 (-B3501 and -B5301) and HLA-A1 (-A2601 and -A0101) had similar antigen presentation results; HLA-A1 (-A3002), HLA-B15 (-B1501), HLA-A3201, and HLA-B58 (-B5701 and -B5801) also had similar antigen presentation results. As for the H2 major cross-reactive clusters, the H2-Ld scores were similar to those of HLA-B7 (-B0702 and -B2101); the H2-Kd scores were similar to those of HLA-A24 (-A2301 and -A2402); H2-Db, H2-Dd, and H2-Kb were grouped in an H2-exclusive manner. However, in the HLA major cross-reactive clusters, the H2-Kk scores were similar to those of HLA-B44 (-B4001, -B4402, and -B4403).

### 3.5. Docking of pMHC Molecules with the Dominant Epitopes

The dominant epitopes with strong affinity, high immunogenicity, and conservation were denoted as “selective” ones, some of which also exhibited pan-MHC-I reactivity. We observed ubiquitous immune reactiveness for HTNV GP-derived 9-mer epitopes among the MHC-I genotypes, superfamilies, clusters, and even across species. In silico validation was simulated by the docking of 11 selective epitopes with human and mouse MHC-I molecules. The binding conformation, docking models, and respective binding scores were obtained through the 10 most important simulations for each of the 9-mer peptides (Figure 2). Lower scores indicated better peptide-MHC docking performance. The results showed that the docking scores of nine epitopes, such as APQCGIKCW and CWFVKSGEW, were lower—in other words, with better docking performance in the HLA-I subtypes than in the mouse H-2 alleles. In contrast, the other two epitopes, RYKSRCYIF and YTYPWHTAK, were more likely to bind to the mouse H-2 allele.

### 3.6. Comparison between the HTNV Strain 76-118 and the Other 147 Variants Based on Nine High-Affinity Segments

We obtained the incidence of variation between HTNV strain 76-118 and the other 147 variants based on nine high-affinity segments (Appendix A). The seven mutations (I222L, I502L, I502V, V996L, I1073M, S1076N, and I1088V) are shown in Figure 3A. The respective frequencies were 39.189% for the mutation I222L, 27.702% for I502L, 9.459% for I502V, 35.135% for V996L, 39.864% for I1073M, 25.676% for S1076N, and 27.027% for I1088V (Figure 3B).

### 3.7. Affinity Differences of HTNV GP Hotspots between 76-118 and Variants

The mutations between HTNV strain 76-118 and the other 147 variants led to the difference in the binding affinity of the corresponding HLA-I and 9-mer epitopes (Appendix A.). Based on a comparison of the differences in binding affinity, we selected the most significant mutation I222L for further analysis (heatmap in Figure 3C and the plane Cartesian coordinate system in Figure 3D). In this aa214-aa230 peptide, the affinity of epitopes AVKGNTYKI and VKGNTYKIF in other variants was generally higher in each subtype of MHC-I, the binding affinity of epitope KIFEQVKKS in other variants was higher in HLA-A2, and the affinity of epitopes NTYKIFEQV, YKIFEQVKK, and IFEQVKSF in HTNV strain 76-118 was generally higher in MHC-I.

Most of the 9-mer epitopes in this peptide showed little difference in their effect on each genotype. However, there was a single mutation in three epitopes leading to eight pHLA-I affinity changes, in which seven pHLA-I bindings were strengthened in variants, but the remaining one was the opposite (Table 4).

### 3.8. Applications HTNV GP-Derived CTL Epitopes by Literature Review

After consulting the previously reported data on HTNV GP-derived CD8^+^ epitopes [7,22,25,26], we summarized those that were proven to activate CTL responses or be presented by MHC-I (Appendix A). Then, cross-alignment was performed with our dominant epitopes in order to assess the HTNV GP CD8^+^ epitopes’ ability to stimulate T-cell immune responses and their application prospect as vaccine candidates. VFV9, VV9, SV9, SL9, FL9, and GP161-169 were among the dominant epitopes we reported corresponding to each MHC-I allele. Subsequently, immunogenicity analyses favored VFV9, SV9, LL9, and GP161-169 as promising targets, while VV9, SL9, FL9, and VI9 were not. In the conservation analysis, only VV9, LL9, VI9, and GP173-181 were conserved as intraspecies, and none of the epitopes were interspecies conservative. The above results confirmed the validity of our study, thus deepening our understanding of further epitope screening for vaccination.

### 3.9. H2d Showed Cross-Immunoreactivity to HLA-A2-Restricted Epitopes in ELISpot Assays

Splenocytes from immunized mice were stimulated with inactivated vaccine and the secretion of IFN-γ was observed. The results (Figure 4) are shown as the average of the spot-forming units (SFUs) per 10^6^ splenocytes, and the ordinate represents the difference between the experimental and the control groups. The assay showed that all three HTNV GP 9-mer peptide epitopes stimulated strong immune responses in Balb/c mice, but only a minor immune response in C57 mice. Epitope VMASLVWPV exhibited the highest IFN-γ secretion, followed by epitope VIGQCIYTI and epitope FLLVLESIL. The *p* value for the data average between the Balb/c group and C57 group was 0.0005 by the Paired *t*-test. Additionally, by using the unpaired *t*-test method, the *p* value for epitope FLLVLESIL was 0.0089, for epitope VMASLVWPV was 0.0045, for epitope VIGQCIYTI was <0.0001, and for the peptides pool was 0.0088. The results indicated that the HLA-A2-restricted epitopes above could also elicit immune responses in Balb/c mice with H2d alleles, but not in C57 mice with H2b alleles, which verified the cross-reactivity in different species and the difference in the HLA reactions of HTNV GP epitopes.

## 4. Discussion

In this study, we identified 11 HTNV GP 9-mer peptides with high affinity to major HLA-I and H-2 superfamilies, evolutionarily conserved and immunogenic, thus named pan-MHC-I selective epitopes. Then, MD simulations with corresponding pMHC-I interactions validated not only the fine postures in antigen presentation, but also the tendencies of cross-species immunoreactivities. Based on the interactions between GP peptides and MHC-I haplotypes, two-way hierarchical cluster analyses revealed similarities among alleles, superfamilies, clusters, and even species. Moreover, 148 HTNV variants yielded six types of major amino acid residue substitutions involving four in nine hotspots, which displayed minimal influences on the general potential of MHC-I superfamily presentation. Studies have shown that HTNV infection can trigger a strong specific CD8^+^ T-cell response in a murine model [27], while the clearance of HTNV infection depends on interferon-γ (IFN-γ) and TNF-α production by functional specific CD8^+^ T cells, suggesting that CD8^+^ T cells play an important role in the clearance of HTNV [28]. CD8^+^ T cells were also necessary to completely remove the virus from infected host cells [29]. Our multidimensional exploration lays a theoretical and technical foundation for the construction of protective CTL epitope vaccines that can activate HTNV-specific population immunity.

Peptides with high binding affinity to MHC-I are considered to be immunogenic. However, these peptides do not always result in high T-cell reactivity [16,30]. In contrast, peptides with low binding affinity do not mean low immunogenicity [31]. Only antigens that are both immunoreactive and immunogenic can elicit immune responses. Thus, immunogenicity analysis has become an integral part of the epitope research process and is widely used in epitope vaccine research for various viruses [32,33,34]. We performed immunogenicity analysis on all nine-peptide epitopes with VaxiJen so that high-affinity, non-immunogenic epitopes could be eliminated.

Studies have shown that, since the discovery of HTNV, it has diverged in rodent populations and undergone geographic isolation and independent evolution [35,36,37]. It has been speculated that the co-evolution of this virus and its host could eventually lead to the emergence of virulent HTNV [38]. Broadly reactive antibodies were proven to elicit extensive recognition and cross-neutralization of Old and New World HTNV [39,40], implying that conserved epitopes of the immune response of interspecies viruses can increase the scope of protective immunity. Therefore, inter- and intra-specific conservation studies have inspected not only the reactiveness in HTNV, but also the cross-reaction to the Old and New World hantavirus. In this study, dominant epitopes were analyzed and then divided into four categories. Epitopes conserved both inter- and intra-species were considered more suitable for epitope vaccine development and immunization studies than other epitopes. This phenomenon can also be seen for SARS-CoV-2, the pathogen of the globe-sweeping pandemic COVID-19 (https://covid19.who.int/, 27 December 2021). It has been shown that HLA-B15:03 possesses the greatest ability to present highly conserved novel coronavirus peptides, which are shared with common human coronaviruses, suggesting that it can achieve cross-protective T-cell-based immunity [41].

Determining the structure of epitope peptide–receptor complexes is necessary for understanding the molecular mechanisms of related biological processes and for vaccine design [19]. The docking scores of pMHC-I simulations associated with 11 selective epitopes and HLA-I and mouse MHC-I demonstrated that, for various alleles across species, the epitopes can be well docked in both humans and mice. Moreover, for different mouse haplotypes, the performance of H2-Ld was better than that of H2-Kb. This once again indicated that BALB/c is more appropriate than C57BL/6 for simulating the responses in humans [23,42]. Intriguingly, nine in eleven selective epitopes showed better docking status with HLA-I than H2, indicating the significance in vaccine development and the immunopathophysiology of HFRS.

Human genetic factors can also influence the susceptibility to transmission and severity of hantavirus-induced diseases [43]. HLA-B46 and HLA-A02 are more frequent in Han Chinese HFRS patients [44,45,46], and HLA-B35 has a significant prevalence in Slovenian HFRS patients [47]. Notably, the pan-MHC-I supertypes in our research contained 27 members of HLA-I superfamilies and six alleles of mouse H2 that extensively cover geography and communities. Meanwhile, our integrated approaches with five forecasting algorithms focused the scope and improved the accuracy of the dominant epitopes. Since epitopes with cross-reactivity between humans and mice included a wide range of MHC molecules of different genera, this seemed to extend beyond MHC-restricted and suggested a genuine advantage during immune responses. For example, the distribution of dominant epitopes in H2d resembled that in the HLA-II superfamily [48]. By hierarchical cluster analysis, we became conscious of the similarity among different HLA-I genotypes in presenting identical epitopes, as well as the similarities between them and mice H2 alleles. It can be speculated from the similar binding scores between H2d/H2k and the wide range of HLA-Is to HTNV GP peptides that, in the absence of humanized HLA-I transgenic mice, hybrids of the BALB/c and C3H strains might be useful surrogates for experimental models.

A variety of hantavirus variants have been found in mainland Asia alone [49,50], and molecular epidemiological surveys of multiple sites have found low homology of hantaviruses [51,52,53]. It was therefore necessary to investigate the variation of antigens. The alignment results showed little intra-species difference in binding affinity between the HTNV 76-118 strain and 147 variants. This provides a basis for the development of epitope vaccines in terms of intra-species conservation. So far, there are multiple vaccine candidates with the potential for conferring long protective immunity against hantaviruses, such as the virus-like particle vaccine, recombinant proteins, the recombinant vector vaccine, and the nucleic acid-based molecular vaccine [54]. Epitope-based vaccines represent a powerful way to induce specific immune responses against selected epitopes, avoiding the side effects in intact antigens. In addition to other considerable advantages, including increasing safety and improving the potency and breadth of vaccines, epitope-based vaccine methods were shown to be successful against various infectious diseases, such as *Neisseria meningitidis* infection, HIV [55], respiratory syncytial virus, and tuberculosis [56]. Therefore, effective T-cell-activating peptide vaccines based on HTNV structural proteins may be a promising approach to disease control [57].

Although HLA-A2 was not assigned to the same cluster as H2d and H2b by the HTNV GP 9-mer peptide bonding affinity profile in the hierarchical clustering, the H2d scores were more similar to those of HLA-A2 than those of H2b when presenting HTNV GP peptides. The amount of IFN-γ secreted by splenocytes from BALB/c mice was significantly greater than that secreted by splenocytes from C57 mice, and it was suggested that HLA-A2-restricted HTNV GP epitopes with pan-MHC-I properties showed a better immune response in H2d than H2b. This was consistent with the result of hierarchical clustering. Additionally, the result of molecule docking confirmed that H2d had a docking score closer to HLA-A2 than H2b, which was verified in the assays. The experimental validation of 11 selective epitopes in a short period of time requires much labor and cost, but after the validation of three HLA-A2-restricted epitopes, the epitopes with multi-MHC-I restriction and cross-reactivity can still be validated in different genotypes, superfamilies, clusters, and species.

Bioinformatics methods used to predict epitopes can be used to assist subsequent immunological experiments and epitope-based vaccine design [23,58,59]. However, the inherent limitations are also not negligible, where hierarchical clustering ignores the interconnectivity and approximation of data. The role of selective epitopes in different genetic backgrounds and in different modes of presentation for antigen processing [60,61] also requires further research and application. Previous studies have reported that the inhibition of molecules LMP7 and BNLF2a attenuated immunoproteasome formation and protein degradation; thus, the MHC-I antigen presentation activity was repressed [62,63], which also reduced the effect of epitopes. Nevertheless, our study presents a state-of-the-art approach to screening dominant epitopes with selective advantages and enhances our understanding of cross-immunity among viruses in different species, providing guidance for the development of epitope vaccines. At the same time, research and discussion of antigen-presenting alteration by variant substitution should also be applied to SARS-CoV-2 variant studies and the ongoing COVID-19 vaccine pursuit.

## Figures and Tables

**Figure 1 vaccines-10-00564-f001:**
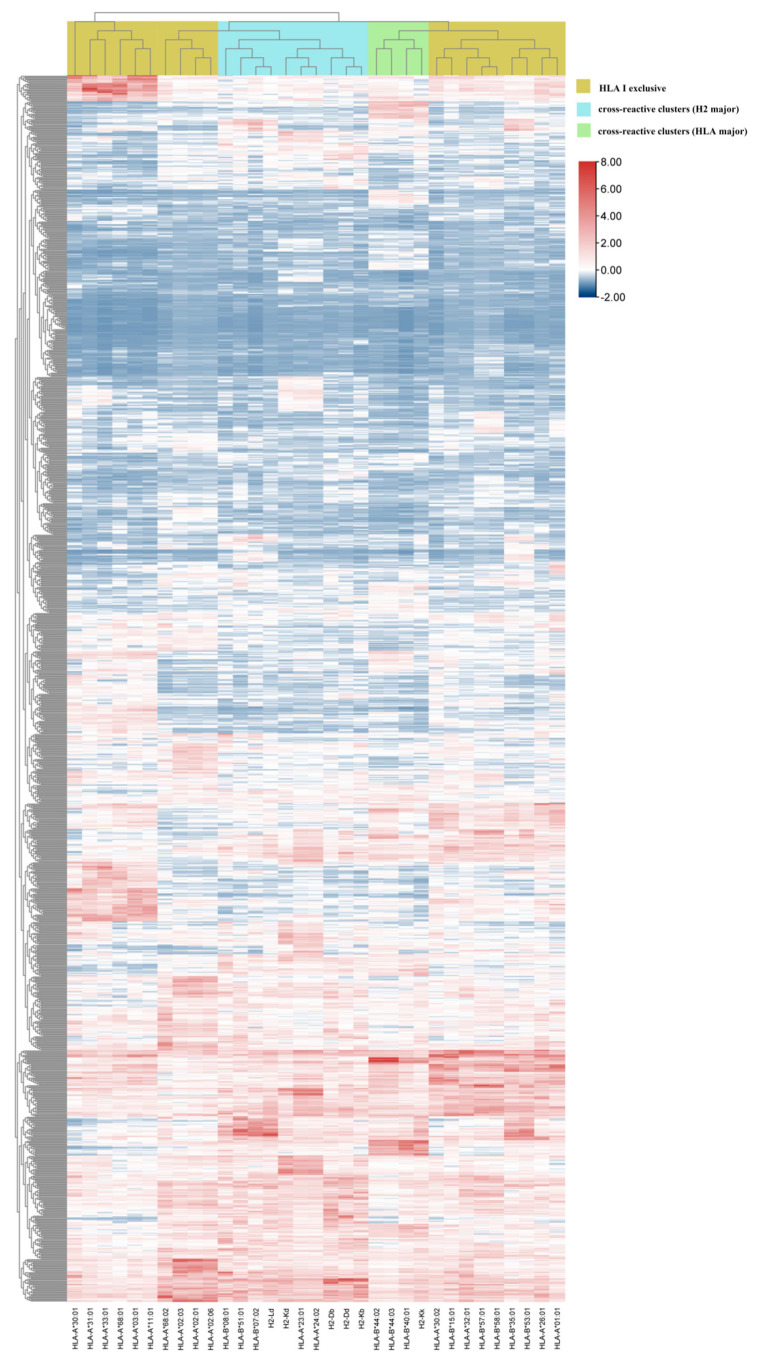
Interaction between HTNV GP 9-mer peptides and pan-MHC-I supertypes. Red represents strong affinity; blue represents weak affinity.

**Figure 2 vaccines-10-00564-f002:**
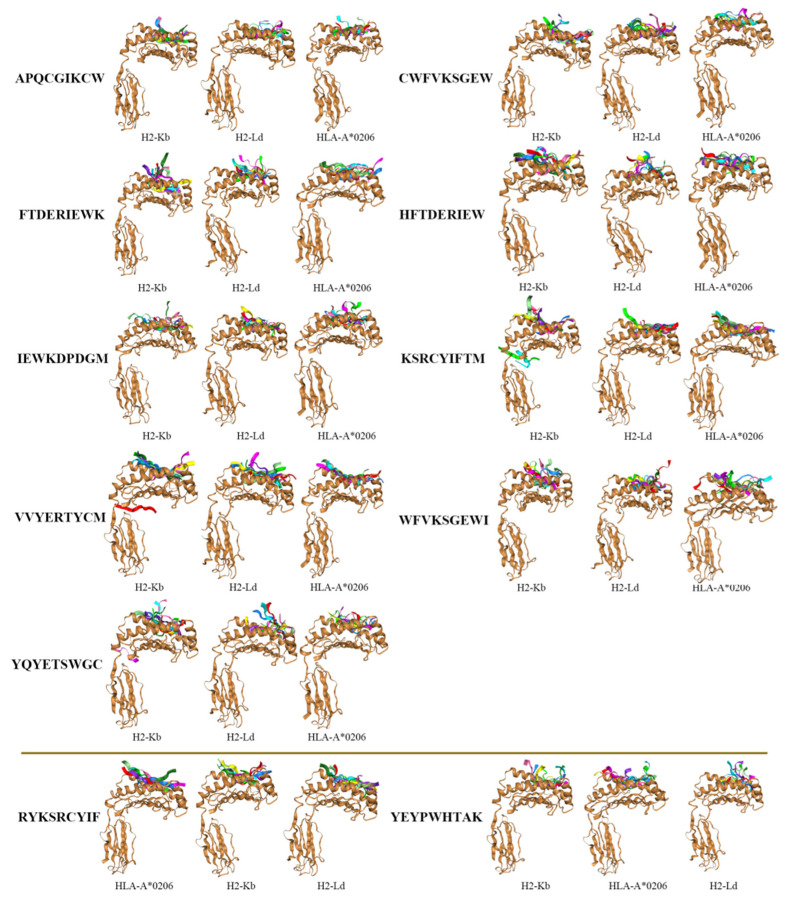
Docking models of selective epitopes with corresponding MHC-I alleles across species. The docking model of each epitope indicates a lower score from left to right. The upper images represent HLA-I-favored binding while the lower images refer to H2 preference.

**Figure 3 vaccines-10-00564-f003:**
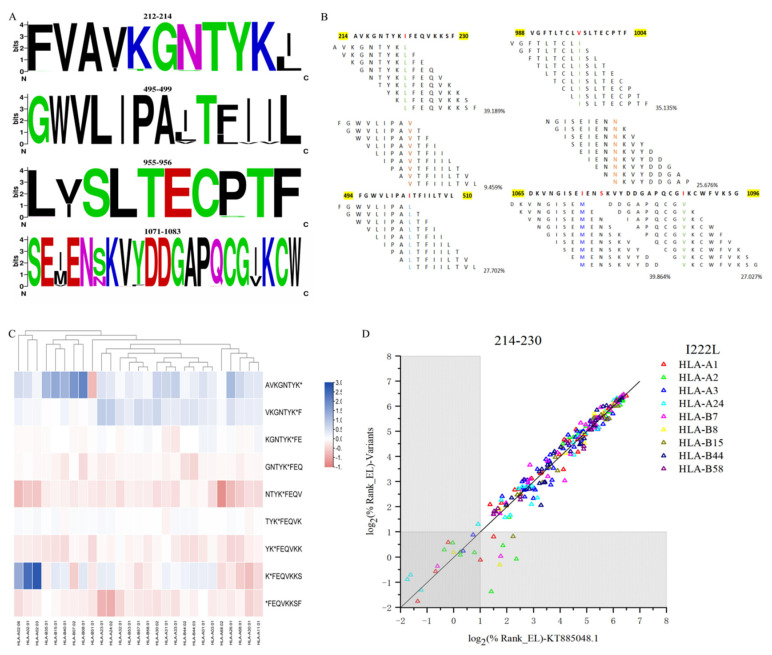
High-affinity segment alignment in HTNV variants. (**A**) Four WebLogo plots with high-frequency mutation sites. (**B**) High-frequency mutation sites and mutation frequency in 148 mutant strains. (**C**) Heatmap of binding affinity differences before and after aa214–aa230 variation. (Red indicates that the mutation causes a decrease in affinity, and blue indicates that the mutation causes an increase in affinity.) (**D**) Scatterplot of binding affinity differences before and after aa214–aa230 variation. (Each point in the scatter plot represents a nonapeptide epitope, and the point in the gray area indicates that the affinity ranking of this epitope is in the top 2%. The closer the point is to the straight line y = x, the closer the affinity between HTNV strain 76-118 and the variants).

**Figure 4 vaccines-10-00564-f004:**
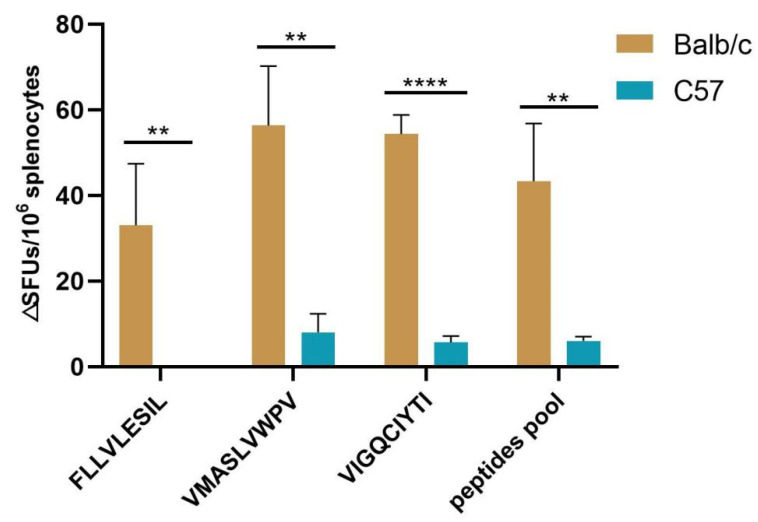
Histogram of three HLA-A2-restricted epitope ELISpot assays. The results are shown as the average of spot-forming units (SFUs) per 10^6^ splenocytes. The ordinate represents the difference between the experimental and the control groups (ΔSUFs/10^6^ splenocytes) and the abscissa represents the different peptides for stimulation. Peptides pool refers to the stimulation after mixing the three peptides. (**, *p* < 0.01; ****, *p* < 0.0001).

**Table 1 vaccines-10-00564-t001:** Numbers of HLA-1-dominant epitopes of HTNV GP.

MHC-I Haplotypes	Prediction Tools	GP Epitopes	GP (Short-Listed)
HLA-A1	IEDB	32	32
	NetMHCpan	32	
	Rankpep	0	
	SMMPMBEC	32	
	SYFPEITHI	10	
HLA-A2	IEDB	38	41
	NetMHCpan	40	
	Rankpep	25	
	SMMPMBEC	28	
	SYFPEITHI	20	
HLA-A3	IEDB	52	57
	NetMHCpan	53	
	Rankpep	35	
	SMMPMBEC	51	
	SYFPEITHI	37	
HLA-A24	IEDB	30	32
	NetMHCpan	30	
	Rankpep	16	
	SMMPMBEC	24	
	SYFPEITHI	16	
HLA-B7	IEDB	40	41
	NetMHCpan	41	
	Rankpep	27	
	SMMPMBEC	35	
	SYFPEITHI	25	
HLA-B8	IEDB	11	11
	NetMHCpan	11	
	Rankpep	0	
	SMMPMBEC	9	
	SYFPEITHI	5	
HLA-B15	IEDB	31	33
	NetMHCpan	32	
	Rankpep	3	
	SMMPMBEC	26	
	SYFPEITHI	21	
HLA-B44	IEDB	26	26
	NetMHCpan	26	
	Rankpep	4	
	SMMPMBEC	23	
	SYFPEITHI	17	
HLA-B58	IEDB	24	25
	NetMHCpan	24	
	Rankpep	19	
	SMMPMBEC	18	
	SYFPEITHI	17	

(GP epitopes are those ranking in the top 2% of each algorithm result; GP (Short-listed) are those that appeared in at least three prediction algorithm results).

**Table 2 vaccines-10-00564-t002:** Numbers of murine MHC-I-dominant epitopes of HTNV GP.

MHC-I Haplotypes	Prediction Tools	GP Epitopes	GP (Short-Listed)
H2-Db	IEDB	17	18
	NetMHCpan	17	
	Rankpep	7	
	SMMPMBEC	12	
	SYFPEITHI	14	
H2-Dd	IEDB	10	12
	NetMHCpan	10	
	Rankpep	8	
	SMMPMBEC	7	
	SYFPEITHI	NA	
H2-Kb	IEDB	15	15
	NetMHCpan	15	
	Rankpep	8	
	SMMPMBEC	11	
	SYFPEITHI	NA	
H2-Kd	IEDB	16	17
	NetMHCpan	16	
	Rankpep	10	
	SMMPMBEC	9	
	SYFPEITHI	13	
H2-Kk	IEDB	12	15
	NetMHCpan	12	
	Rankpep	5	
	SMMPMBEC	8	
	SYFPEITHI	8	
H2-Ld	IEDB	12	15
	NetMHCpan	13	
	Rankpep	7	
	SMMPMBEC	7	
	SYFPEITHI	6	

(GP epitopes are those ranking in the top 2% of each algorithm result; GP (Short-listed) are those that appeared in at least three prediction algorithm results).

**Table 3 vaccines-10-00564-t003:** Conservation of MHC-I-restricted dominant epitopes of HTNV GPs.

MHC-I Haplotypes	Interspecies- Intraspecies-	Interspecies- Intraspecies+	Interspecies+ Intraspecies-	Interspecies+ Intraspecies+
H2-Db	16	2	0	0
H2-Dd	10	2	0	0
H2-Kb	12	2	0	1
H2-Kd	11	5	0	1
H2-Kk	13	2	0	0
H2-Ld	10	3	0	2
HLA-A1	22	9	0	1
HLA-A2	34	6	0	1
HLA-A3	40	14	0	3
HLA-A24	17	9	0	6
HLA-B7	27	10	1	3
HLA-B8	5	4	0	2
HLA-B15	21	9	0	1
HLA-B44	19	5	0	2
HLA-B58	15	9	1	0

**Table 4 vaccines-10-00564-t004:** Epitopes that generated changes and their HLA molecules.

Amino Acid Number	KT885048.1	Variants	Dominant in KT885048.1	Dominant in Variants	HLA-I Genotype
214–222	AVKGNTYKI	AVKGNTYKL	No	Yes	HLA-B07:02
			No	Yes	HLA-B08:01
			No	Yes	HLA-B15:01
			No	Yes	HLA-A30:02
218–226	NTYKIFEQV	NTYKLFEQV	Yes	No	HLA-A32:01
221–229	KIFEQVKKS	KLFEQVKKS	No	Yes	HLA-A02:01
			No	Yes	HLA-A02:03
			No	Yes	HLA-A02:06

## Data Availability

Data are contained within the article or Appendix A.

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
