# Peer review of "Comparative Immunoreactivity Analyses of Hantaan Virus Glycoprotein-Derived MHC-I Epitopes in Vaccination"

_vaccines, 2022, doi:10.3390/vaccines10040564_

Round 1

Reviewer 1 Report

In the submitted manuscript the authors assessed potential immunogenic peptides derived from the GP of HFRS using commonly available in silico analyses. Determining conservation in HFRS genotypes and counterselecting for autoreactivity they identify several peptides with potential for vaccine usability. 

The readability of the manuscript is marred because of poor English. It would allow the reader to more clearly assess the rationale of the performed analyses and the conclusions if the manuscript were to be edited by a native English speaker or such service provider. 

The main limitation of the presented study is the lack of validation of immunogenicity of the proposed peptides. The authors should perform CD8T activation assays using some of the proposed peptides. 

Given that the GP protein is not cytoplasmic but produced in the ER the production of peptides by the immunoproteasome and presentation on MHCI would reasonably be thought to be limited. As such the relevance of using GP derived peptides in vaccines would be small. The authors should discuss this caveat. 

Author Response

After carefully reading all the comments that reviewer have put forward, we find them very truthful and helpful to our study, and we will elucidate all the points one by one and deeply hope that our responds will answer all the questions, thus improve the quality of the manuscript.

Appended answers to the reviewer’s comments:

In the submitted manuscript the authors assessed potential immunogenic peptides derived from the GP of HFRS using commonly available in silico analyses. Determining conservation in HFRS genotypes and counterselecting for autoreactivity they identify several peptides with potential for vaccine usability.

A#: We appreciate the reviewer for the objective commentary.

The readability of the manuscript is marred because of poor English. It would allow the reader to more clearly assess the rationale of the performed analyses and the conclusions if the manuscript were to be edited by a native English speaker or such service provider.

A#: We thank the reviewer for the constructive comments. We have revised the manuscript by MDPI language service and sincerely hope that the language problem could be solved. Please refer to the revised manuscript.

The main limitation of the presented study is the lack of validation of immunogenicity of the proposed peptides. The authors should perform CD8T activation assays using some of the proposed peptides.

A#: We appreciated this comment and thought it would be great if all the prediction could be validated. However, it could not be able to accomplished because of the relative long term of oligo-peptide de novo synthesize in the Lunar New Year and the temporary quarantine in domestic China. We acquired 3 of already synthesized HTNV GP-derived HLA-A2 dominant 9-mer epitopes from Dr.Tang and performed preliminary validation. Details was narrated below and could also be referred in the revised manuscript.

The inactivated bivalent HFRS vaccine (HANPUWEI®, purified HTNV and SEOV from Hamster Kidney Cells) was only commercialized in China and used to prime the instant response in Balb/c and C57 mice. On the 6th day after immunization, mice cellular responses specific to three HLA-A2 restricted HTNV GP 9-mer peptides were validated by enzyme-linked immunospot (Elispot) assay. The results showed the stronger immune response in Balb/c mice of H2d restricted and mere response in C57 mice of H2b restricted. According to the prediction result, although HLA-A2 was not assigned with H2d and H2b in the same cluster by the profile of HTNV GP 9-mer peptides bonding affinity in the hierarchical clustering, the H2d scored more similar to HLA-A2 than H2b when presenting HTNV GP peptides. The amount of IFN-γ secreted by splenocytes from BALB/c mice was significantly more than that secreted by splenocytes from C57 mice and it suggested that HLA-A2 restricted HTNV GP epitopes with pan-MHC-I property performed better immune response in H2d than H2b. This is consistent with the result of hierarchical clustering. Also the result of molecule docking confirmed that H2d had a docking score closer to HLA-A2 than H2b, which was verified in the assays. Since experimental validation of 11 selective epitopes in a short period of time requires much labor and cost, but after validation of three HLA-A2-restricted epitopes, the epitopes with multi-MHC-I restriction and the cross-reactivity can still be validated in different genotypes, superfamilies, clusters and species.

Given that the GP protein is not cytoplasmic but produced in the ER the production of peptides by the immunoproteasome and presentation on MHCI would reasonably be thought to be limited. As such the relevance of using GP derived peptides in vaccines would be small. The authors should discuss this caveat.

A#: Thanks for the valuable comments. As the glycoprotein expressed in the ER, it may result in a limited role in the processing of submissions, but this has not been reported in HTNV. Previous studies have reported that inhibition of molecules LMP7 and BNLF2a attenuated immunoproteasome formation and protein degradation, thus the MHC-I antigen presentation activity was repressed. In addition, Hantavirus Gn has been reported inducing the macro autophagy, which suggests that the removal of Gn takes the MHC-II pathway. It has been reported that there are strongly responsive epitopes on GP, suggesting the feasibility of the MHC-I pathway. Therefore, how the MHC-I presentation pathway is regulated facing up within viral glycoproteins should be further studied.  It has been revised as “The role of selective epitopes in different genetic backgrounds and in different modes of presentation for antigen processing also require further research and application. Previous studies have reported that inhibition of molecules LMP7 and BNLF2a attenuated immunoproteasome formation and protein degradation, thus the MHC-I antigen presentation activity was repressed, which also reduced the effect of epitopes.

Reviewer 2 Report

The authors in manuscript entitle "Comparative analyses of pan-MHC-I selective epitope comprehend HTNV glycoprotein immunoreactivity in vaccination" investigated different Hantaan virus GP-derived 9-mer dominant epitopes using molecular docking software to find the best candidate for vaccine development and activate CD8+T cell response. From their screening the authors manage to find 11 HTNV GP 9-mer peptides with high-affinity to major HLA-I and H-2 superfamilies, evolutionarily conserved and immunogenic. 

The study design probably and the authors use appropriate methods to perform the study. The results part explain nicely in details and easy to understand. And finally the discussion part written and supported with other research finding.

I have few comments for the authors to improve their manuscript;

  1. In the title of manuscript please use the full name of virus not the short term for virus
  2. In scientific writing better not to use terms such as "WE study..." or "Our data show..."
  3. I suggest the authors send their manuscript for English proof reading  

Author Response

First of all, we would like to appreciate reviewer for his/her positive commentary. After carefully reading all the comments that reviewer have put forward, we gratefully find them very truthful and helpful to our study, and we are willing to address all the issues one by one and deeply hope that our responds will answer all the comments, thus improve the quality of the manuscript.

Appended answers to the reviewer’s comments:

  1. In the title of manuscript please use the full name of virus not the short term for virus.

A#:We thank the reviewer for the reminder. And we correct this in the revised version. Since another reviewer has also concerned the title logic, it has been revised as “Comparative immunoreactivity analyses of Hantaan virus glycoprotein-derived MHC-I epitopes in vaccination”

  1. In scientific writing better not to use terms such as "WE study..." or "Our data show..."

A#: We thank the reviewer for the suggestion. Considering that it helps make the objective expression, we fully adopted it and revised in the manuscript.

  1. I suggest the authors send their manuscript for English proof reading.

A#: Thanks for reviewer’s supportive comments. And we have done this by language editing service and please refer to the revised manuscript.

Reviewer 3 Report

Reviewers comments vaccines-1552114:

Sun et al., present an interesting and valuable study examining the identification of HTNV glycoproteins epitopes to elicit CD8+ T cell responses. The content of this manuscript is great. However, in its current form it is difficult to digest the information contained in the manuscript, and to identify the key messages including but no confined to the objective, key findings, outcomes, and future applicability of this work, as well as further studies required to confirm the findings. The authors also fail to clearly define the caveats of this study and what could be done to close these gaps. Therefore, major rewriting and reformatting of this manuscript is required to improve the clarity and flow of this paper. After which I believe, this paper will be significantly improved.

General and Introduction:

  1. Check grammar, spelling and sentence structure. At present the flow of the manuscript and the clarity of the key messages are disrupted.
  2. Rephrase title as it doesn’t currently make sense and doesn’t clearly communicate the content of the paper
  3. In vitro, In vivo, In silco etc. should italicized
  4. Line 66 “HTNV Gn-derived” please correct as presumably this should be Gp-derived?
  5. Clearly outline the objective of this study at the end of the introduction as this is currently unclear.

Results:

  1. Ensure that peptide binding alogorithms are adequately referenced, as there are publications describing each algorithm.
  2. Lines 185 – 189 “Generally the binding strength …”. Please reword as at present I am unsure of that the loss of numbers mean, is each range for a specific algorithm or is the hierarchy for which regions are hotspots for peptide epitopes. Please specify.
  3. Lines 1910195 – Please specify if this is a hierarchy for the HLA in which the most epitopes have been predicted.
  4. Tables 1 and 2 – what is the difference between the GP epitopes and GP(Short listed) columns. Please add a footnote at the bottom of the table to explain.
  5. Table 3 – Please make this clearer as at present I am confused about what the columns in this table represent.
  6. Is their any functional data to support immunogenicity of the identified epitopes and therefore, their utility in epitope screening for vaccination. If not then please address in the discussion.

Author Response

After carefully reading all the comments that reviewer have put forward, we gratefully find them very truthful and helpful to our study, and we will elucidate all the points one by one and deeply hope that our responds will answer all the questions, thus improve the quality of the manuscript.

Appended answers to the reviewer’s comments:

Further comments to this commentary are shown below.

Sun et al., present an interesting and valuable study examining the identification of HTNV glycoproteins epitopes to elicit CD8+ T cell responses. The content of this manuscript is great. However, in its current form it is difficult to digest the information contained in the manuscript, and to identify the key messages including but no confined to the objective, key findings, outcomes, and future applicability of this work, as well as further studies required to confirm the findings. The authors also fail to clearly define the caveats of this study and what could be done to close these gaps. Therefore, major rewriting and reformatting of this manuscript is required to improve the clarity and flow of this paper. After which I believe, this paper will be significantly improved.

A#: We thank the reviewer for his/her supportive comments on our manuscript. As for the language aspect of this manuscript, we have revised the manuscript by MDPI language service and sincerely hope that the information contained in the manuscript could digest more easily. Also, this study has some limitations and shortcomings, and we have recently performed experiments to validate the partial findings and narrow the gaps as much as possible. Meanwhile, we have modified the flow of the manuscript to make the logic more clear. Please refer to the revised manuscript.

  • Check grammar, spelling and sentence structure. At present the flow of the manuscript and the clarity of the key messages are disrupted.

A#: Thanks for reviewer’s comments. We have revised the manuscript by MDPI language service and sincerely hope that the language problem could be solved. Also, we have modified the flow of the manuscript to make the logic more clear. Please refer to the revised manuscript.

  • Rephrase title as it doesn’t currently make sense and doesn’t clearly communicate the content of the paper.

A#:We thank the reviewer for the reminder. And we rephrased the title in the revised version. It has been revised as “Comparative immunoreactivity analyses of Hantaan virus glycoprotein-derived MHC-I epitopes in vaccination”. The change and amendment do not affect the main finding of this paper. Please refer to the revised manuscript.

  • In vitro, In vivo, In silco etc. should italicized.

A#: Thanks for reviewer’s comments. And we have corrected the problem. Please refer to the revised manuscript.

  • Line 66 “HTNV Gn-derived” please correct as presumably this should be Gp-derived?

A#: Thanks for reviewer’s meticulous comments. And we have corrected the problem. Please refer to the revised manuscript.

  • Clearly outline the objective of this study at the end of the introduction as this is currently unclear.

A#: Thanks for reviewer’s valuable comments. Indeed, the "Multidimensional exploration" in manuscript was not clearly expressed. And it has been revised as "We investigated HTNV CTL epitopes from immune-affinity, immunogenicity, conservation and molecular docking. Cross-reactivity of HTNV CTL epitopes was investigated by hierarchical cluster analysis. The multidimensional exploration laid a theoretical and technical foundation for the development of protective CTL epitope vaccines that can activate HTNV specific population immunity in the context of pan-MHC immunology."

  • Ensure that peptide binding alogorithms are adequately referenced, as there are publications describing each algorithm.

A#: Thanks for reviewer’s meticulous comments. We have noticed the problem and corrected it. Please refer to the revised manuscript.

  • Lines 185 – 189 “Generally the binding strength …”. Please reword as at present I am unsure of that the loss of numbers mean, is each range for a specific algorithm or is the hierarchy for which regions are hotspots for peptide epitopes. Please specify.

A#:Thanks for reviewer’s meticulous comments. Based on the result of NetMHCPan4.1 and Supplementary Figure 1, the mentioned numbers are chosen as the high or low affinity peptides. The loss of (not mentioned) numbers mean the regions that are mixed with the high and low affinity epitopes or the regions that the affinity is generally moderate.

  • Lines 1910195 – Please specify if this is a hierarchy for the HLA in which the most epitopes have been predicted.

A#: Thanks for reviewer’s question. We used all the HTNV GP 9-mer peptides for HLA hierarchy clustering.

  • Tables 1 and 2 – what is the difference between the GP epitopes and GP(Short listed) columns. Please add a footnote at the bottom of the table to explain.

A#: Thanks for reviewer’s strict comment. We have added the footnote at the bottom of the table to explain. Please refer to the revised manuscript.

  • Table 3 – Please make this clearer as at present I am confused about what the columns in this table represent.

A#: We are glad to make complementary explanation for the reviewer’s confusion. And we apologize for not explaining the meaning of "+" "-" in the manuscript. And it has been revised as ""+" means the epitopes were conservative, on the contrary, "-" means the epitopes were not conservative." The columns represent conservation status for the dominant epitopes of each MHC-I haplotype. ’Interspecies- Intraspecies-’ means the epitope counts which are neither interspecies nor intraspecies conservation. ’Interspecies+ Intraspecies-’ means the epitope counts which are only interspecies but not intraspecies conservation. ’Interspecies- Intraspecies+’ means the epitope counts which are only intraspecies but not interspecies conservation. ’Interspecies+ Intraspecies+’ means the epitope counts which are both intraspecies and interspecies conservation.

  • Is their any functional data to support immunogenicity of the identified epitopes and therefore, their utility in epitope screening for vaccination. If not then please address in the discussion.

A#: We appreciated this comment and thought it would be great if all the prediction could be validated. However, it could not be able to accomplished because of the relative long term of oligo-peptide de novo synthesize since the up-coming Lunar New Year and the temporary quarantine in domestic China. We acquired 3 of already synthesized HTNV GP-derived HLA-A2 dominant 9-mer epitopes from Dr.Tang and performed finite validation. Details was narrated below and could also be referred in the revised manuscript.

The inactivated bivalent HFRS vaccine (HANPUWEI®, purified HTNV and SEOV from Hamster Kidney Cells) was only commercialized in China and used to prime the instant response in Balb/c and C57 mice. On the 6th day after immunization, mice cellular responses specific to three HLA-A2 restricted HTNV GP 9-mer peptides were validated by enzyme-linked immunospot (Elispot) assay. The results showed the stronger immune response in Balb/c mice of H2d restricted and mere response in C57 mice of H2b restricted. According to the prediction result, although HLA-A2 was not assigned with H2d and H2b in the same cluster by the profile of HTNV GP 9-mer peptides bonding affinity in the hierarchical clustering, the H2d scored more similar to HLA-A2 than H2b when presenting HTNV GP peptides. The amount of IFN-γ secreted by splenocytes from BALB/c mice was significantly more than that secreted by splenocytes from C57 mice and it suggested that HLA-A2 restricted HTNV GP epitopes with pan-MHC-I property performed better immune response in H2d than H2b. This is consistent with the result of hierarchical clustering. Also the result of molecule docking confirmed that H2d had a docking score closer to HLA-A2 than H2b, which was verified in the assays. Since experimental validation of 11 selective epitopes in a short period of time requires much labor and cost, but after validation of three HLA-A2-restricted epitopes, the epitopes with multi-MHC-I restriction and the cross-reactivity can still be validated in different genotypes, superfamilies, clusters and species.

Round 2

Reviewer 1 Report

line 369: "The value among the three epitopes was 0.0005. " what does this mean? Were the increases in spots statistically different from background? 

Which test was used to determine this? Also, since no activation was seen in the C57 mice for the first peptide it is not likely that this was statistically significant. 

Line 479: "Previous studies have reported that inhibition of molecules LMP7 and BNLF2a attenuated immunoproteasome formation". Please clarify how or by what these are inhibited.

line 365: "106 splenocytes" the 6 should be upper case

Legend of Fig. 4 should be expanded to describe the experiment done and what is represented, the figure should be interpretable by itself and the legend. 

Author Response

Response to the reviewer #1’s comments:

line 369: "The value among the three epitopes was 0.0005. " what does this mean? Were the increases in spots statistically different from background?

A#: We appreciate the reviewer for the objective commentary. We apologize for not explaining this. The value in the sentence referred to the P value. We used the Paired t test to validate the difference between the Balb/c group and C57 group. And the output P value was 0.0005 which meant the data was significant. Also, we conducted further studies on data validity. By using the method of unpaired t test, the P value in epitope FLLVLESIL is 0.0089, in epitope VMASLVWPV is 0.0045, in epitope VIGQCIYTI is <0.0001, and in the peptides pool is 0.0088. It has been revised as “The P value of data average between the Balb/c group and C57 group was 0.0005 by Paired t test. Also, by using the method of unpaired t test, the P value for epitope FLLVLESIL is 0.0089, for epitope VMASLVWPV is 0.0045, for epitope VIGQCIYTI is <0.0001, and for the peptides pool is 0.0088.” At the same time, we marked the test results on the figure. Please refer to the revised manuscript. As for the second problem, the ordinate in figure 4 represents the difference between the experimental and the control group (ΔSUFs/106 splenocytes). The background spots numbers of Balb/c control are 47, 47, 35, and 30, and those of C57 control are 23, 10, 6, and 8.

Which test was used to determine this? Also, since no activation was seen in the C57 mice for the first peptide it is not likely that this was statistically significant.

A#: We thank the reviewer for the constructive comments. We used the Paired t test to validate the data average between the Balb/c group and C57 group. And the output P value was 0.0005, which confirmed the statistic significance. Also, we conducted further studies on data validity. By using the method of unpaired t test, the P value for epitope FLLVLESIL is 0.0089, for epitope VMASLVWPV is 0.0045, for epitope VIGQCIYTI is <0.0001, and for the peptides pool is 0.0088. As for the second problem, since the number of spots after stimulation with the first peptide of C57 was less than that of the control group, and its difference was negative, we did not show it in Fig 4.

Line 479: "Previous studies have reported that inhibition of molecules LMP7 and BNLF2a attenuated immunoproteasome formation". Please clarify how or by what these are inhibited.

A#: Thanks for reviewer’s meticulous comments. A study on the HIV virus found that HIV-1 negative regulatory factor (Nef) counteracts host immunity, particularly the MHC-I response. Nef interacts with LMP7 on the endoplasmic reticulum (ER), downregulating the incorporation of LMP7 into immunoproteasome and thereby attenuating its formation. This study identified a distinct mechanism by which Nef facilitates immune evasion via suppressing the function of immunoproteasome and MHC-I [1]. Another study on Epstein-Barr virus found that immediately after initiation of the lytic cycle, BNLF2a prevents the import of peptides into the ER by transporter associated with antigen processing (TAP), blocking the binding of both peptides and ATP to the transporter complex. Also, Viral-IL10 can cause a reduction in mRNA levels of TAP1 and bli/LMP2, a subunit of the immunoproteasome. These EBV lytic proteins result in highly effective interference with CD8(+) and CD4(+) T cell surveillance, thereby providing the virus with a window for undisturbed generation of viral progeny [2]. Since these are less relevant to our research topic, this content was not added to the manuscript.

line 365: "106 splenocytes" the 6 should be upper case

A#: Thanks for reviewer’s meticulous comments. And we have corrected the problem. Please refer to the revised manuscript.

Legend of Fig. 4 should be expanded to describe the experiment done and what is represented, the figure should be interpretable by itself and the legend.

A#: Thanks for the valuable comments. It has been revised as “The results are shown as the average of spot-forming units (SFUs) per 106 splenocytes. The ordinate represents the difference between the experimental and the control group (ΔSUFs/106 splenocytes) and abscissa represents different peptides for stimulation. Peptides pool refers to the stimulation after mixing the three peptides. ” Please refer to the revised manuscript.

References

[1] Yang Y, Liu W, Hu D, Su R, Ji M, Huang Y, Shereen MA, Xu X, Luo Z, Zhang Q, Liu F, Wu K, Liu Y, Wu J. HIV-1 Nef Interacts with LMP7 To Attenuate Immunoproteasome Formation and Major Histocompatibility Complex Class I Antigen Presentation. mBio. 2020 Oct 27;11(5):e02221-19. doi: 10.1128/mBio.02221-19. PMID: 33109760; PMCID: PMC7593969.

[2] Ressing ME, Horst D, Griffin BD, Tellam J, Zuo J, Khanna R, Rowe M, Wiertz EJ. Epstein-Barr virus evasion of CD8(+) and CD4(+) T cell immunity via concerted actions of multiple gene products. Semin Cancer Biol. 2008 Dec;18(6):397-408. doi: 10.1016/j.semcancer.2008.10.008. Epub 2008 Oct 25. PMID: 18977445.

Reviewer 3 Report

I commend the authors for taking on my comments. There have been significant improvements in the manuscript. No additional edits are required from my perspective.

Author Response

Reply to reviewer 3#

Thanks for your commendation. And we appreciate your help to improve our manuscript.